# Preparation and Characterization of Ibuprofen Containing Nano-Embedded-Microparticles for Pulmonary Delivery

**DOI:** 10.3390/pharmaceutics15020545

**Published:** 2023-02-06

**Authors:** Petra Party, Márk László Klement, Piroska Szabó-Révész, Rita Ambrus

**Affiliations:** Interdisciplinary Excellence Centre, Institute of Pharmaceutical Technology and Regulatory Affairs, University of Szeged, Eötvös Street 6, 6720 Szeged, Hungary

**Keywords:** ultrasound, nanoprecipitation, ibuprofen, nano spray-drying, dry powder inhaler, cystic fibrosis

## Abstract

A fatal hereditary condition, cystic fibrosis (CF) causes severe lung problems. Ibuprofen (IBU), a non-steroidal anti-inflammatory drug, slows the progression of disease without causing significant side effects. Considering the poor water-solubility of the drug, IBU nanoparticles are beneficial for local pulmonary administration. We aimed to formulate a carrier-free dry powder inhaler containing nanosized IBU. We combined high-performance ultra-sonication and nano spray-drying. IBU was dissolved in ethyl acetate; after that, it was sonicated into a polyvinyl alcohol solution, where it precipitated as nanoparticles. Mannitol and leucine were added when producing dry particles using nano-spray drying. The following investigations were implemented: dynamic light scattering, laser diffraction, surface tension measurement, scanning electron microscopy, X-ray powder diffraction, differential scanning calorimetry, Fourier-transform infrared spectroscopy, in vitro dissolution test, and in vitro aerodynamic assessment (Andersen Cascade Impactor). The particle diameter of the IBU was in the nano range. The spray-dried particles showed a spherical morphology. The drug release was rapid in artificial lung media. The products represented large fine particle fractions and proper aerodynamic diameters. We successfully created an inhalable powder, containing nano-sized IBU. Along with the exceptional aerodynamic performance, the ideal particle size, shape, and drug-release profile might offer a ground-breaking local therapy for CF.

## 1. Introduction

Cystic fibrosis (CF) is an autosomal recessive disease affecting mucus and sweat-producing cells in multiple organs. The respiratory system is the most severely affected, leading to death in 90% of patients [1]. A mutation in the cystic fibrosis transmembrane conductance regulator (CFTR) gene results in a modification of the activity of chloride and sodium channels. CFTR is present in the apical surface of the epithelial cells lining the respiratory tract, biliary tree, intestines, vas deferens, sweat ducts, and pancreatic ducts [2]. The defective CFTR causes thick mucus obstructing small airways, and thus providing an optimal environment for bacterial growth and infiltration of lung tissue by neutrophils [3,4].

Respiratory system complications include bronchiectasis caused by inflammation, chronic infections leading to pneumonia, chronic rhinosinusitis, nasal polyposis, hemoptysis, pneumothorax, and eventually respiratory failure. Respiratory exacerbations include airway symptoms (e.g., increased coughing and sputum production, change in sputum appearance, and shortness of breath), and systemic complications (e.g., fever, weight loss, and fatigue). Each exacerbation episode increases the risk of permanent loss of lung function, which will lead to death [1,5]. Digestive problems include nutritional deficiencies, such as absorption of fat and fat-soluble vitamins, and usually manifest in diabetes. Hepatic dysfunction, gallstones, intestinal obstruction, intussusception, small intestine bacterial overgrowth, and distal intestinal obstruction syndrome may also evolve. The imbalance of minerals in the blood leads to dehydration, arrhythmias, fatigue, and weakness. Other complications may include infertility, osteoporosis, and mental health issues [1,2].

The main goal of CF therapy is to maintain lung function, reduce the appearance of exacerbations and improve the patient’s quality of life [5]. For respiratory problems, mucolytic agents are useful. Reducing viscoelasticity and removing thick, sticky mucus from the lungs by inhaling hypertonic saline solution, dornase-alfa, and mannitol are recommended [6]. Additionally, exercise and physiotherapy can also help the patients. Medications to treat the airway symptoms in CF are designed to improve the clearance of mucus from the lungs and to treat chronic infections and inflammation. Another challenge is that patients with CF have an increased volume of distribution and faster rate of clearance, therefore requiring larger doses of drugs with shorter intervals between doses. Part of the day-to-day treatment is to control lung infections (e.g., Staphylococcus aureus, Haemophilus influenzae, Pseudomonas aeruginosa) and prevent acute exacerbations with the help of oral or inhaled antibiotics (e.g., azithromycin, tobramycin, aztreonam, colistin, levofloxacin). To treat acute respiratory complications, oral, inhaled, or intravenous antibiotics are also administered [1,7]. Anti-inflammatory therapy is one of the new strategies for treating CF pulmonary disease. To moderate airway inflammation non-steroidal anti-inflammatory drugs (NSAIDs), systemic and inhaled steroids, some antimicrobial agents that show anti-inflammatory effects, and antioxidants are used [3]. Chronic inflammation damages the airway wall, leading to bronchiectasis and progressive decline in pulmonary function. The challenge in developing an anti-inflammatory medication for CF lies in reducing harmful inflammation without unpleasant side effects [8,9].

NSAIDs are widely used in therapy due to their analgesic, antipyretic, and anti-inflammatory effects [4]. Among them, ibuprofen (IBU) is the only drug approved for chronic use in CF. IBU inhibits the migration, adhesion, and aggregation of the leukocytes, and decreases the release of lysosomal enzymes. It maintains optimal body weight and improves FEV1 (Forced Expiratory Volume in one second) [10]. The long-term use of high-dose oral IBU (20–30 mg/kg/dose twice daily, targeting a peak plasma drug concentration of 50–100 µg/mL) can significantly slow down the disease progression in CF patients between 6 and 17 years of age, without unacceptable adverse effects [11,12]. IBU therapy is cost effective and has extensive history of pediatric use. The disadvantages are that it can cause gastrointestinal bleeding and renal dysfunction [4,10]. IBU also acts synergistically with antibiotics and sodium chloride, and thus might play a multifunctional role in the treatment of CF [13,14].

Inhalation delivery of IBU can be more efficient for the treatment of CF as the lungs are the desired targets for anti-inflammatory effects [15]. Local therapy administers drugs directly to the lungs, with limited absorption into the systemic circulation, therefore minimizing the possible side effects [16]. A large surface area is available for administration with a dense vasculature, which provides for rapid onset of action. Degradation of drugs by gastrointestinal enzymes and first-pass metabolism in the liver does not occur [17]. Therefore, it will require a reduced fraction of the dose for an equivalent therapeutic effect in the lungs compared to the oral route [18]. In the case of IBU, an estimated maximum of 300 mg dose of IBU per day for pulmonary drug delivery is required for CF treatment, in contrast to the maximum of 3 g per day of the oral dose [4,9].

The poor water-solubility of ibuprofen causes a certain difficulty in the formulation of the drug. Furthermore, attention should be paid to the model drug’s low melting point (75–78 °C). The application of nanoparticles could be a solution because they are beneficial formulations for Class II drugs of the Biopharmaceutics Classification System (BCS), where the dissolution rate is the rate-limiting step for absorption. A reduction in particle size can increase the dissolution rate as a result of a higher specific surface area [19]. Therefore, the preparation of inhalable IBU nanoparticles with increased dissolution may be used as an alternative formulation approach for the targeted delivery of IBU to the lungs. [20,21,22]. Literature exists about the micronization and nanonization of IBU, but mostly to enhance its water-solubility and basically for oral application [23,24,25,26,27]. Inhalable formulations for local treatment have been published about micro-sized IBU-containing particles [15,28] and only one describes nanoparticle agglomerates [29].

Our research work aimed to investigate the feasibility of the preparation of an ibuprofen nanosuspension to achieve rapid dissolution. From the nanosuspension, we planned to develop inhalable microparticles to achieve high lung deposition. Therefore, a combination of various preparation methods was used. Additives were applied to investigate the possibility of changing the formulations’ physico-chemical and aerodynamic properties and improve the dissolution rate of the drug. Considering the different nanosizing techniques, ultrasound-sonication was chosen. It is a reproducible, cost-effective, and scalable way of preparing nanosuspensions [30,31,32]. The solidification of the nanosuspensions combines the advantages of liquid nanosuspensions (i.e., enhanced dissolution and solubility) with the benefits of solid formulations (i.e., stability, easier applicability) producing nanoparticle agglomerates. Spray-drying is a popular process from an industrial perspective because it is also cost and time efficient and scalable [33,34,35,36]. Preparation of nanosuspensions by sonication, combining nano spray-drying to produce dry powders, is suggested as a formulation approach for “nano-in-micro” particles with enhanced dissolution and aerosolization efficiency [37,38,39,40,41]. The novelty of this work was to formulate suitable nano-embedded-microparticles containing IBU for pulmonary delivery, using modern preparation techniques. Inhalable dry powder inhalers containing nanosized IBU could provide a potential local treatment for CF.

## 2. Materials and Methods

### 2.1. Materials

The active ingredient was ibuprofen (IBU) (Sanofi, Veresegyház, Hungary). Ethyl acetate (EtOAc) (Molar Chemicals Kft, Halásztelek, Hungary) was used as a solvent. Poly-vinyl-alcohol 4-98 (PVA), (Aldrich Chemistry, Darmstadt, Germany) was used as a stabilizer. D-Mannitol (MAN) (Molar Chemicals Kft, Halásztelek, Hungary) and L-leucine (LEU), (AppliChem GmbH, Darmstadt, Germany) were applied as excipients to improve aerodynamic properties.

### 2.2. Methods

#### 2.2.1. Spray-Ultrasound Associated Solvent Diffusion-Based Nanoprecipitation

To select the appropriate organic solvent and stabilizer, preliminary experiments were performed. In the case of the most promising formulation, 40 mg of IBU was solved in 10 g EtOAc. Then, the solution was mixed with an aqueous 0.2% (*w*/*v*) PVA solution. The initial temperature was between 17.10 ± 0.65 °C. A high-performance ultrasound homogenizer (UP 200ST ultrasonic device, Hielscher Ultrasonics GmbH, Teltow, Germany) was used for 2 min. The following settings were applied during the method: pump rate: 25 mL/min, power: 50%, cycle: 50%, amplitude: 100%. A coarse emulsion was obtained. Purified water was added to dilute the emulsion with the ultrasonic homogenizer, using the previous parameters. As a result of the sonication, drug particles were precipitated. The temperature increased to 33.25 ± 0.19 °C. During homogenization, the drug diffused from the organic solvent and converted the droplets into solid particles. This precipitation resulted in a nanosuspension [42,43].

#### 2.2.2. Nano Spray-Drying

The novelty of the preparation technology of the nano spray-drying lies in the vibration mesh spray technology to form small droplets, producing powders even in the submicron particle size range, with narrow distribution and high yields. The gentle technique is acceptable for the IBU, which has a low melting point [34,35]. Dry powder inhalers were formulated from the previously described nanosuspension via nano spray-drying (Büchi Nano Spray Dryer B-90 HP, Büchi, Flawil, Switzerland). MAN and LEU were applied as additives to improve the aerodynamic properties of the final samples [44,45]. Mannitol is also helpful in cystic fibrosis; thanks to its osmotic effect, it can dilute the thick mucus [6,46]. The ratios was based on data from the literature [29]. Drying parameters were as follows, considering the low melting point of the drug: temperature: 70 °C, spray: 100%, pump: 20%, flow rate: 120 l/min. The spray mesh was a membrane with 4.0 μm sized holes. The final composition of the spray-dried samples and the yield of the nano spray-drying are shown in Table 1. The IBU content of the spray-dried samples was determined. The results were close to the theoretical values. Physical mixtures were also prepared from the initial materials to compare their properties with the spray-dried samples.

### 2.3. Particle Size Analysis

#### 2.3.1. Dynamic Light Scattering (DLS)

Using a Malvern Zetasizer Nano ZS, the average hydrodynamic diameter (Z-average), polydispersity index (PDI), and potential were measured (Malvern Instruments, Worcestershire, United Kingdom). Pure water was applied to dilute the nanosuspension, and purified water was also used to dispense the spray-dried formulations before they were tested in folded capillary cells at 25 °C. The refractive index of IBU was adjusted to 1.550. The experiment was carried out three times.

#### 2.3.2. Laser Diffraction

Laser diffraction was used to determine the particle size, the particle size distribution, and the specific surface area of our samples (Malvern Mastersizer Scirocco 2000, Malvern Instruments Ltd., Worcestershire, UK). The refractive index of IBU was set to 1.550. The solid particles were observed using the dry dispersion equipment. A dispersion air pressure of 3.0 bar and 75% vibration feed were used. Each sample was tested in triplicate. The values of D [0.1] (10% of the volume distribution is below this value), D [0.5] (50% of the volume distribution is below this value), and D [0.9] (90% of the volume distribution is below this value) were used to describe the particle size distribution (PSD). The particle size distribution provided Span values; the higher the Span value, the broader the distribution. The PSD values were used to calculate the specific surface area (SSA). The calculations were performed with the presumption that the particles were spherical. SSA data predicted the dissolution properties of the products.

### 2.4. Surface Tension Measurement

The pendant drop technique was used with an OCA 20 equipment (Dataphysics Instrument GmbH, Filderstadt, Germany) for the measurements of the interfacial tensions between liquid and gas for the different fluids (PVA solution and the IBU nanosuspensions). These measurements are based on a balance between interfacial forces keeping the droplet attached to the needle and gravitational ones pulling the droplet down. A drop of the fluid was formed through an injection needle inside a, and then the integrated camera (Dataphysics Instrument GmbH, Filderstadt, Germany) focused and captured the drop images collected at 25 °C. The geometry of the drop was recorded using integrated software (SCA 20, Dataphysics Instrument GmbH, Filderstadt, Germany). The software calculated the surface tension of the sample by using the Young–Laplace equation. For the estimations, the density values of the samples were previously measured and adjusted. For each experiment, ten subsequent images were collected and the average surface tension was calculated.

### 2.5. Investigation of Morphology

Scanning electron microscopy (SEM) was used to examine the morphology of the spray-dried powders (Hitachi S4700, Hitachi Scientific Ltd., Tokyo, Japan). The parameters were the following: 10 kV high voltage, 10 mA amperage, and 1.3–13.1 mPa air pressure. A high vacuum evaporator and argon atmosphere were administered to make the sputter-coated samples conductive with gold-palladium (Bio-Rad SC 502, VG Microtech, Uckfield, UK). Based on the SEM records, particle diameter investigations were carried out using ImageJ image analyzer software (https://imagej.net/ij/index.html, accessed on 5 February 2023).

### 2.6. Structural Analysis

#### 2.6.1. X-ray Powder Diffraction (XRPD)

Structural analysis was implemented with the help of the BRUKER D8 Advance X-ray diffractometer (Bruker AXS GmbH, Karlsruhe, Germany). The device recorded the spectra of the initial materials, the physical mixtures, and the spray-dried samples. The radiation source was Cu Kλ1 radiation (λ = 1.5406 Å). The parameters of the analysis were the following: Cu target, Ni filter, 40 kV voltage, 40 mA current, time constant 0.1°/min, angular step 0.010° throughout 3–40°. The data were evaluated with DIFFRACT^plus^ EVA software (Bruker AXS GmbH, Karlsruhe, Germany).

#### 2.6.2. Differential Scanning Calorimetry (DSC)

DSC measurements were executed with a Mettler Toledo DSC 821e thermal analysis system with the STAR^e^ thermal analysis program version 9.1 (Mettler Inc., Schwerzenbach, Switzerland). Approximately 2–5 mg of the samples was measured in the temperature range between 25 °C and 300 °C. The heating rate was 10 °C/min. Argon was the carrier gas at a flow rate of 10 l/h during the investigations.

#### 2.6.3. Fourier-Transform Infrared Spectroscopy (FT-IR)

The interactions between IBU and the excipients were investigated by the AVATAR 330 FT-IR spectrometer (Thermo Nicolet, Thermo Fisher Scientific Inc., Waltham, MA, USA). The sample was homogenized with 150 mg of dry KBr in an achate mortar, and the mixture was pressed to create pastilles using a Specac^®^ hydraulic press (Specac, Inc., Orpington UK) with 10-ton pressing force. The infrared spectra were recorded between 4000 and 400 cm^−1^, at an optical resolution of 4 cm^−1^.

### 2.7. In Vitro Dissolution Measurement

A modified method was used to investigate the in vitro drug release of IBU. The medium was 20 mL of simulated lung fluid, the pH of which was 7.4 ± 0.1. The artificial media contained 0.68 g/L NaCl, 2.27 g/L NaHCO_3_, 0.02 g/L CaCl_2_, 0.1391 g/L NaH_2_PO_4_, 0.37 g/L glycine, and 5.56 mL/L 0.1 M H_2_SO_4_. The magnetic stirrer was set to 200 rpm. The amount of the dissolved API was determined in real time at the wavelength of 222 nm, for 60 min at 37 °C with a spectrophotometric sonda (FDP-7UV200-VAR, Avaspec-ULS2048-USB2, Avantes, Apeldoorn, The Netherlands). Three parallel measurements were performed.

### 2.8. In Vitro Aerodynamic Investigation

The in vitro aerodynamic properties of the nano spray-dried formulations were examined with an Andersen Cascade Impactor (ACI), (Apparatus D, Copley Scientific Ltd., Nottingham, UK). A 60 l/min inhalation flow rate was chosen (High-capacity Pump Model HCP5, Critical Flow Controller Model TPK, Copley Scientific Ltd., Nottingham, UK). The actual flow rate through the impactor was determined by a mass flow meter (Flow Meter Model DFM 2000, Copley Scientific Ltd., Nottingham, UK). The inhalation time was 4 s. With a 4 l inhalation volume, the setting simulates a typical breathing pattern of a person. Single-dose devices of Breezhaler^®^ (Novartis International AG, Basel, Switzerland) and size 3, Ezeeflo™ hydroxypropyl methylcellulose capsules (ACG-Associated Capsules Pvt. Ltd., Mumbai, India) were used. Following the inhalation procedure, the apparatus, the capsules, the induction port, the plates, and the filter were washed with a combination of 90 + 10 *v*/*v*% methanol and pH 7.4 phosphate buffer to collect and dissolve the IBU that had been deposited on them. By using UV/Vis spectrophotometry (ATI-UNICAM UV/VIS Spectrophotometer, Cambridge, UK) at a wavelength of 222 nm, the drug was quantified. Using Inhalytix^TM^ (Copley Scientific LTD., Nottingham, UK) data analysis software, which is approved to determine aerodynamic parameters from the particle size distribution, the in vitro aerodynamic properties were assessed. Fine particle fraction (FPF) and median mass aerodynamic diameter (MMAD) are the most widely used values. FPF is defined as the percentage of the mass of the drug-containing of particles with an aerodynamic diameter of under 5 μm divided by the emitted dose (ED) of the formulations. The ED is the released amount of drug from the DPI device. MMAD is the particle diameter, which is formed during inhalation.

### 2.9. Statistical Analysis

Statistical analysis was performed using GraphPad Prism 8.0.1. software (GraphPad Software, San Diego, CA, USA). Student’s t-test was used to determine the statistical significance. Changes were considered statistically significant at *p* < 0.05. All reported data are means ± standard deviation (SD).

## 3. Results

### 3.1. Particle Size Measurements

#### 3.1.1. Dynamic Light Scattering (DLS)

The results of the DLS investigations are presented in Table 2. In case of the IBU suspension, the particle size of the drug decreased into the nano size range (178.72 ± 7.73 nm) after the ultrasound-associated precipitation. The average mesh space range of CF sputum is about 60−300 nm; therefore, the prepared IBU nanoparticles may go through the dense fiber mesh of CF mucus. The particle size lower than 200 nm and the negative ζ-potential will probably lead to high biocompatibility in human bronchial epithelium cells [47]. According to the ζ-potential results (−20.91 ± 2.10 mV), the nanosuspension was described as a stable suspension system. The stability results supported that; thus, the particle size and zeta potential remained persistent for a period of 7 days [48,49]. The diameters of the dispersed nano spray-dried particles, which were produced from the IBU nanosuspension, were between 300–400 nm. It predicts that the dry particles will disintegrate into nanoparticles when they deposit the lung fluid. Particles with a mean particle diameter between 200 and 500 nm are beneficial to reaching increased dissolution and cellular uptake [50]. The negative ζ-potential values of the spray-dried formulations mean that the systems are more degradable and therefore less retentive in the airways. It is advantageous in CF because the particles will not induce further inflammation, infection, and fibrosis [51].

#### 3.1.2. Laser Diffraction

The particle size of the spray-dried particles was acceptable for pulmonary delivery since the D [0.5] values were in the 1–5 µm range (Table 3) [52]. The geometric diameter of IBU_PVA_spd was around 2.13 µm. The geometric size of the spray-dried particles was reduced and the particle size distribution was improved when MAN and LEU were incorporated into the formulations. The particle size distributions were reported in Appendix A (see Appendix A). In comparison to the raw materials, the specific surface area (SSA) values increased, which indicated that the dissolution profile will be enhanced.

### 3.2. Surface Tension Investigation

According to the literature, the initial surface tensions of the two phases are the following: purified water 72.75 ± 0.36 mN/m and EtAc 23.97± 0.10 [53,54,55]. The surface tension of the polymeric stabilizer (0.2% (*w*/*v*) PVA solution) was 78.03 ± 0.165 mN/m. Rapid nanoparticle formation was explained as a process due to differences in surface tension. Since the aqueous phase with a high surface tension pulls more strongly on the surrounding liquid than the organic phase solvent with a low surface tension. This phenomenon causes interfacial turbulence and thermal inequalities in the system, leading to the continuous swirls of solvent at the interface of both liquids. Spreading is seen as a result of the solvents’ mutual miscibility; the solvent flows away from regions of low surface tension and the polymer orients aggregate on the oil surface, creating nanoparticles [56]. The surface tension value of the IBU containing nanosuspension was decreased to 56.44 ± 1.73 mN/m. Adding MAN and LEU further reduced the surface tension of the nanosuspension (48.10 ± 0.74 mN/m). Fluids with lower surface tension generate slightly smaller droplets during nano spray-drying [36], which explains the particle size results of laser diffraction measurements.

### 3.3. Investigation of Morphology

The SEM pictures of the spray-dried formulations presented spherical morphology (Figure 1). The particle shape is a result of nano spray-drying [35]. PVA coated the IBU particles, promoting their separation from each other; thus, no aggregation could be detected [57]. Applying MAN also helped to form spherical particles. IBU_PVA_MAN_LEU_spd showed a rougher surface because LEU has a large Peclet number, which leads to a corrugated surface after spray-drying. The more wrinkled particles will guarantee dispersibility during aerosolization [58,59,60,61]. The observed particle sizes in the SEM records were 671.41 ± 132.38 nm (IBU_PVA_spd) and 561.41 ± 111.62 nm (IBU_PVA_MAN_LEU_spd). Application of the additives decreased the particle size according to DLS and SEM investigations.

### 3.4. Structural Analysis

#### 3.4.1. X-ray Powder Diffraction (XRPD)

The XRPD spectra of the raw materials demonstrated, that IBU, MAN, and LEU had a crystalline structure and PVA showed no crystallinity. In the case of the spray-dried products, the intensities of the characteristic peaks decreased (Figure 2). Ultrasound sonication and nano spray-drying reduced the original crystallinity of IBU to zero. In the case of IBU_PVA_MAN_LEU_spd small peaks were observed, which indicated the presence of partially crystalline MAN. The nano-spray-drying method resulted in the partial amorphization of MAN and LEU.

#### 3.4.2. Differential Scanning Calorimetry (DSC)

On the DSC curves, IBU had a sharper peak at 78.52 °C. LEU showed an endothermic peak at 295.78 °C and MAN at 163.90 °C. PVA had no endothermic peak. The results correlated with the literature data. The curves reflect their melting point and crystalline structure. After the combined preparation method, the DSC curves showed a lower endothermic peak at 54.86 °C in the case of IBU_PVA_spd. This indicates a decrease in the crystallinity, and the residual crystals melted at a lower temperature because of the particle size reduction of the drug. IBU_PVA_MAN_LEU_spd had a peak at 159.03 °C, which detected the partial amorphization of MAN (Figure 3). The DSC results correlated with the XRPD measurements regarding the crystalline structure of the spray-dried formulations.

#### 3.4.3. Fourier-Transform Infrared Spectroscopy (FT-IR)

FT-IR spectral analysis was performed to study the possibility of molecular interactions between IBU and the excipients (PVA, MAN, LEU). To identify the changes occurring during nanonization, the FT-IR spectra of the physical mixtures and the spray-dried formulations were compared (Figure 4). The presence of IBU is disclosed by the peaks at 1721 and 2955 cm^−1^ on the spectra of physical mixtures. They have been attributed to the stretching of O-H and C=O bonds. These bonds belong to the propanoic acid -COOH group present in the ibuprofen molecule. The spectrum is very complex, especially in the region of lower wavenumbers, due to the presence of the other components of the pharmaceutical formulation. Despite this, the above peaks are so well defined that they can be considered the fingerprints of IBU in FT-IR spectra [62,63]. Around 2950 cm^−1^, MAN and LEU also showed characteristic peaks. PVA was recognizable in the spray-dried samples at 3250 cm^−1^ [59,64,65]. It was found that the intensity of the characteristic peaks of IBU decreased during the preparation process, but there was no considerable difference observed in the characteristic peaks, with a slight shift in the values of the peaks in the FT-IR spectra of spray-dried formulations of ibuprofen compared to pure drugs (Table 4). This indicated that there was no interaction between drugs and excipients [66].

### 3.5. In Vitro Dissolution Measurement

The results of the in vitro dissolution test confirmed our aim; the prepared samples showed immediate release in artificial lung media. Thanks to the improved surface area and amorphization of the IBU, the dissolution succeeded rapidly (Figure 5). The total amount of the nano-sized drug was released from the spray-dried samples within the first 2 min (Figure 6). Compared to the prepared formulations, the physical mixtures reached total drug release only after 30–60 min. Rapid dissolution is important for efficient drug delivery in CF therapy, where the properties of the mucus are abnormal [67].

### 3.6. In Vitro Aerodynamic Investigation

The distribution of the products on the stages of the ACI is shown in Figure 7. Approximately half of the IBU_PVA was deposited on the induction port, which was modeling the upper airways. The application of MAN and LEU improved the aerosolization properties of the samples. Large amounts of deposited material on the third and fourth stages were observed, which is equivalent to the bronchial area. The calculated aerodynamic results by Inhalytix™ software are presented in Table 5. The samples had an MMAD between 2–4 μm, which is suitable for the pulmonary delivery guidelines [68]. The inhaled particles could target the related bronchial area of the lung in CF. IBU_PVA_MAN_LEU_spd showed a great FPF (71%) result [69]. This result exceeded the FPF values of the commercially available Breezhaler^®^ DPIs [70]. In addition, the MAN and LEU-containing product is promising for pulmonary administration in the therapy of CF.

## 4. Discussion

In this article, nanosized IBU-containing carrier-free dry powder inhalers were successfully formulated. Nanosuspension of the poorly water-soluble IBU stabilized with PVA was prepared by a high-performance ultrasound sonication-assisted solvent diffusion method. The particle diameter was lower than 200 nm (179 nm). The nanosuspension was further nano spray-dried with or without the addition of excipients (MAN, LEU). The method resulted in nanosized IBU containing inhalable dry powders. The low melting point of the drug was considered during nano spray-drying. The SEM images presented an advantageous spherical particle shape. The composites contained stabilizer polymer (PVA), which prevented the aggregation of the nanoparticles. The formulation resulted in acceptable morphology and particle size properties for pulmonary delivery. Two different analytical methods (XRPD, DSC) indicated the amorphization of the IBU. The formulations were released rapidly during the in vitro dissolution test thanks to the increased surface area and the amorphous form of the API. According to the Andersen Cascade Impactor measurements, the aerodynamic particle diameters of the samples (2–4 μm) were in the pulmonary required micrometric range. Moreover, the IBU_PVA_MAN_LEU_spd showed a higher fine particle fraction (71%), than the commercial products. In addition, the nanosized IBU-containing microparticles could be efficient in the treatment of CF.

## Figures and Tables

**Figure 1 pharmaceutics-15-00545-f001:**
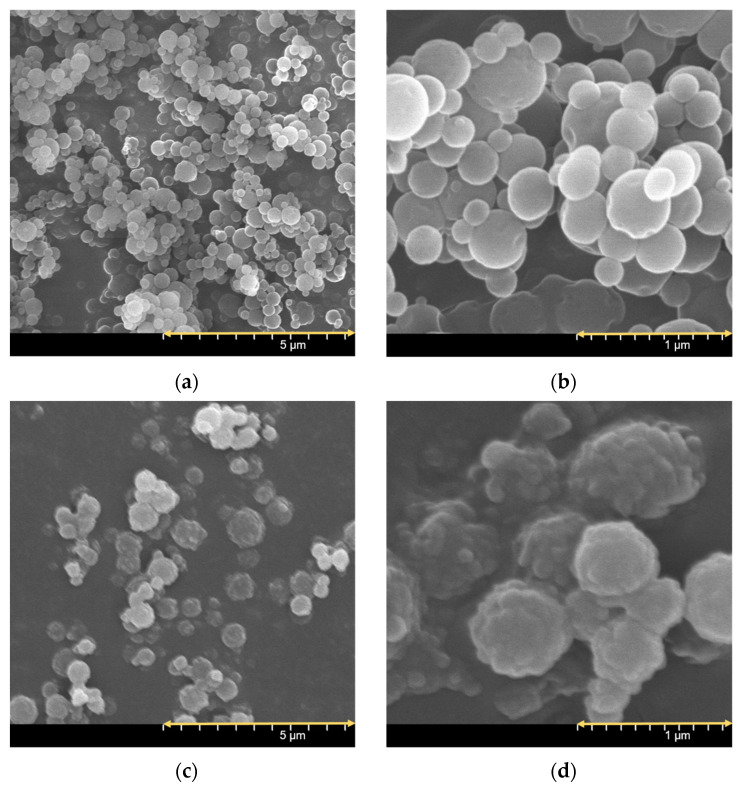
SEM pictures of the spray-dried sample at different magnifications: IBU_PVA_spd (**a**,**b**), IBU_PVA_MAN_LEU_spd (**c**,**d**).

**Figure 2 pharmaceutics-15-00545-f002:**
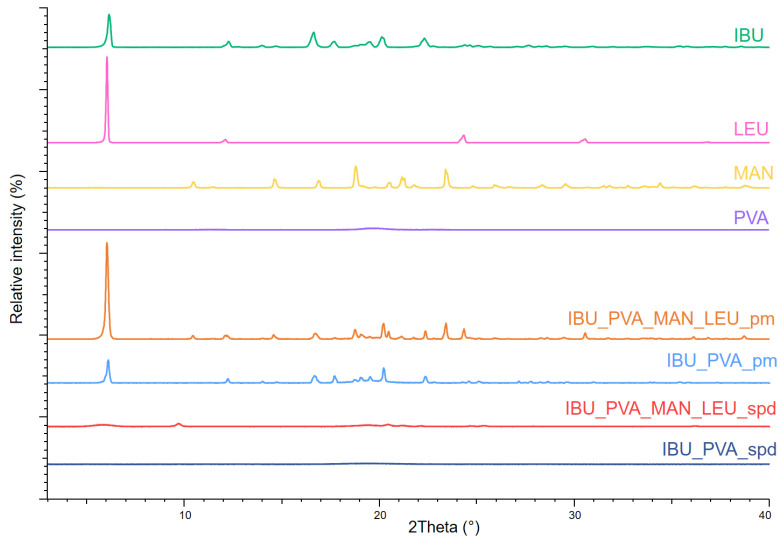
XRPD spectra of the raw materials, the physical mixtures, and the spray-dried samples.

**Figure 3 pharmaceutics-15-00545-f003:**
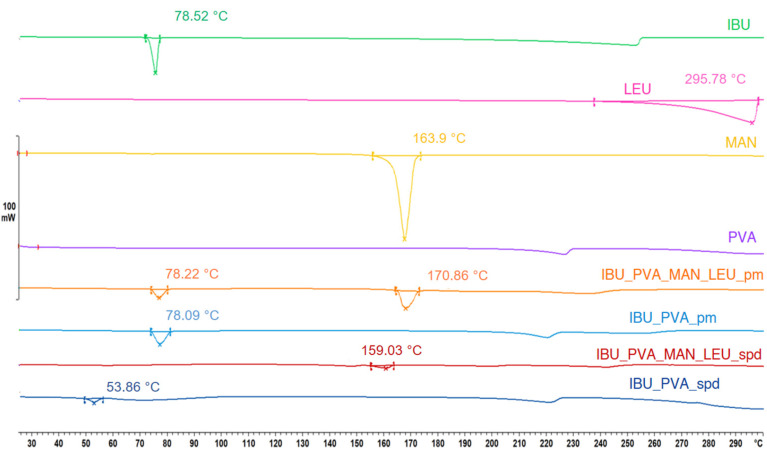
DSC results of the raw materials, the physical mixtures, and the spray-dried samples.

**Figure 4 pharmaceutics-15-00545-f004:**
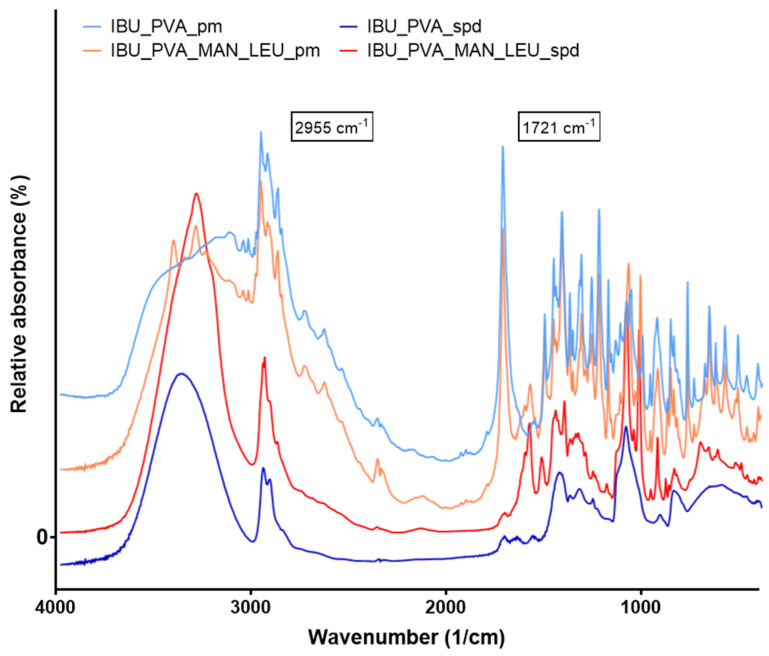
FT-IR spectra of the physical mixtures and the spray-dried samples.

**Figure 5 pharmaceutics-15-00545-f005:**
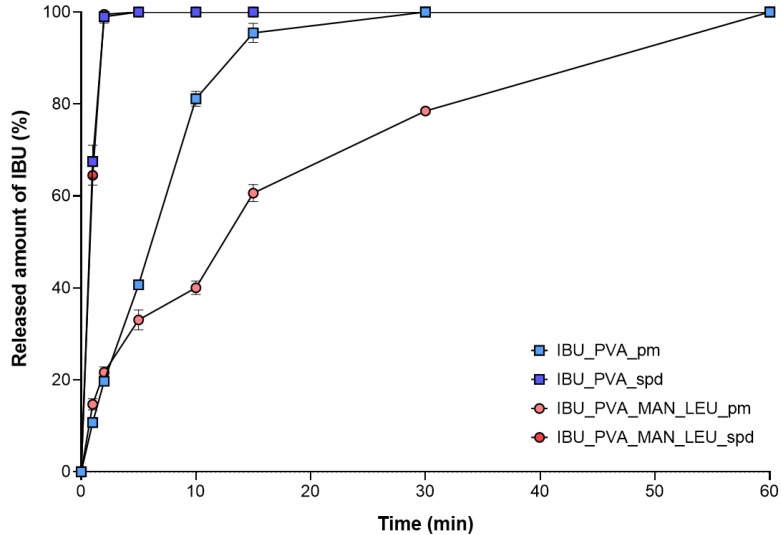
In vitro dissolution results of the physical mixtures and the spray-dried samples. Data are means ± SD (*n* = 3 independent measurements).

**Figure 6 pharmaceutics-15-00545-f006:**
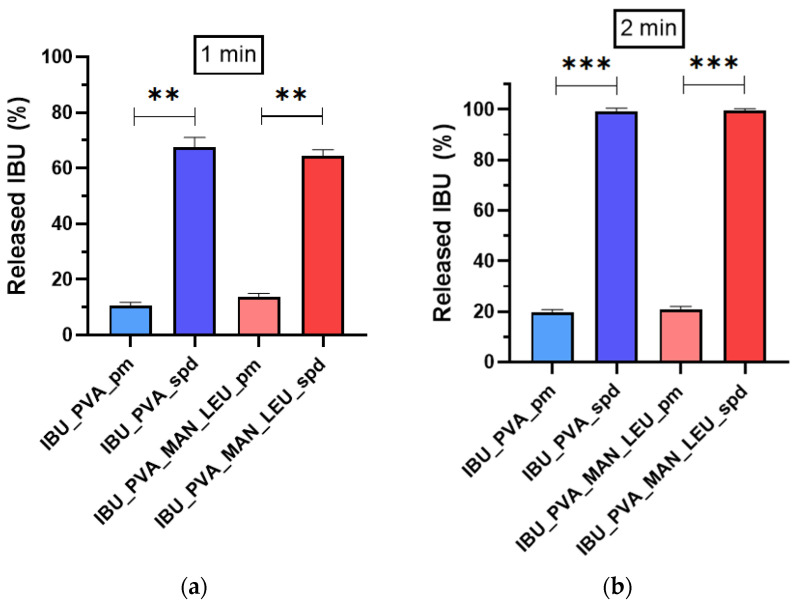
In vitro dissolution results of the physical mixtures and the spray-dried samples at 1 min (**a**) and at 2 min (**b**). Data are means ± SD (*n* = 3 independent measurements). Level of significance: ** *p* < 0.01), *** *p* < 0.001).

**Figure 7 pharmaceutics-15-00545-f007:**
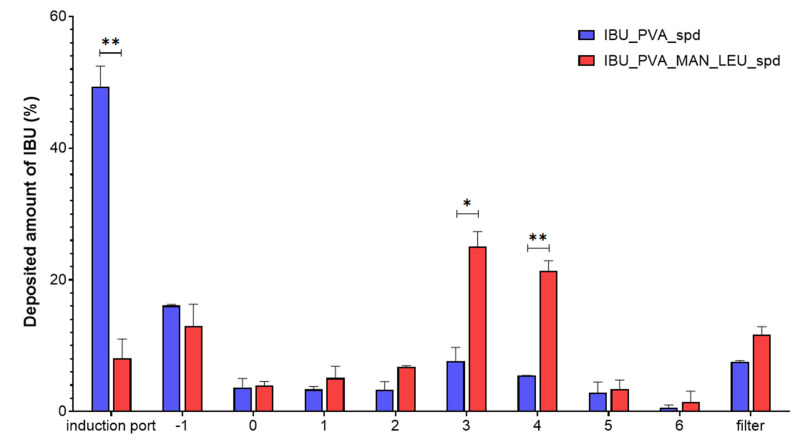
In vitro aerodynamic distribution of the spray-dried samples. Data are means ± SD (*n* = 3 independent measurements). Level of significance: * *p* < 0.05, ** *p* < 0.01.

**Table 1 pharmaceutics-15-00545-t001:** Ratio between the ingredients of the spray-dried samples and physical mixtures.

Sample	IBU (mg)	PVA (mg)	MAN(mg)	LEU (mg)	Yield (%)
IBU_PVA_spd	40	80	-	-	50.00 ± 0.02
IBU_PVA_pm	40	80	-	-	-
IBU_PVA_MAN_LEU_spd	40	80	40	20	46.33 ± 0.08
IBU_PVA_MAN_LEU_pm	40	80	40	20	-

**Table 2 pharmaceutics-15-00545-t002:** DLS investigation results of the nanosuspension and the spray-dried formulations.

Sample	d (nm)	PDI	ζ-Potential (mV)
IBU_PVA_suspension	178.72 ± 7.73	0.40 ± 0.04	−20.91 ± 2.10
IBU_PVA_spd	377.99 ± 35.54	0.43 ± 0.07	−10.80 ± 2.81
IBU_PVA_MAN_LEU_spd	335.33 ± 8.96	0.83 ± 0.17	−21.23 ± 6.77

Data are means ± SD (*n* = 3 independent measurements).

**Table 3 pharmaceutics-15-00545-t003:** Laser diffraction results of the initial API and the spray-dried formulations.

Sample	D [0.5] (μm)	Span	SSA (m^2^/g)
IBU_raw	29.33 ± 0.94	0.74 ± 0.08	0.25 ± 0.04
IBU_PVA_spd	2.13 ± 0.10	11.89 ± 2.67	4.93 ± 0.11
IBU_PVA_MAN_LEU_spd	1.57 ± 0.04	5.55 ± 0.64	5.36 ± 0.20

Data are means ± SD (*n* = 3 independent measurements).

**Table 4 pharmaceutics-15-00545-t004:** Characteristic peaks of ibuprofen and its spray-dried formulations on FT-IR spectra.

Peak	Reported	Observed	IBU_PVA_spd	IBU_PVA_MAN_LEU_spd
C=O stretching	1721	1721	1713	1711
Bonded –OH stretching	2955	2955	2942	2936

**Table 5 pharmaceutics-15-00545-t005:** Aerodynamic results of the spray-dried formulations.

Sample	MMAD (μm)	FPF (%)
IBU_PVA_spd	3.50 ± 0.35	28.35 ± 1.52
IBU_PVA_MAN_LEU_spd	2.24 ± 0.05	70.65 ± 2.47

Data are means ± SD (*n* = 3 independent measurements).

## Data Availability

Not applicable.

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
