# Peer review of "Preparation and Characterization of Ibuprofen Containing Nano-Embedded-Microparticles for Pulmonary Delivery"

_pharmaceutics, 2023, doi:10.3390/pharmaceutics15020545_

Round 1

Reviewer 1 Report

pharmaceutics-2165150

Preparation and characterization of ibuprofen containing nano-embedded-microparticles for pulmonary delivery

The manuscript by Party et al. described the preparation of IBU microparticles by spray-drying for pulmonary delivery. The authors presented sufficient data to demonstrate a successful nano-in-micro spray-dried powder containing IBU for pulmonary delivery. However, there are several issues to consider, as follows.

1. The Introduction part should be re-arranged and modified. In paragraphs 1-5, citations are only included at the end of the paragraphs. Inhalable IBU has been previously reported in some studies; please include and discuss these systems and then highlight the novelty and contribution of this study.

2. Table 1 is vague. Please modify it to clarify the values presented.

3. Section 2.2.1. Spray-ultrasound associated solvent diffusion-based nanoprecipitation: was the organic solvent evaporated? Did the authors determine the residue solvent?

4. Section 2.2.2. Nano spray-drying: please convert the pump 20% to L/h or mL/min.

5. The authors should include a figure to present the size distribution of the spray-dried powders.

6. Figure 3: Please explain the peak at 53.86 °C of IBU_PVA_spd.

7. The authors should include the yield of the spray-drying process.

8. The authors should determine the IBU content in the spray-dried powders and compare them with theoretical values,

9. The authors should clarify the reasons for choosing MAN and LEU as well as their concentrations (1 and 0.5%, respectively) in the spray drying.

10. Please clarify which one is correct, D[0.5] or D[0,5].

Author Response

  1. The Introduction part should be re-arranged and modified. In paragraphs 1-5, citations are only included at the end of the paragraphs. Inhalable IBU has been previously reported in some studies; please include and discuss these systems and then highlight the novelty and contribution of this study.

Thank you for the comment. The introduction is modified.

The poor-water solubility of ibuprofen causes a certain difficulty in the formula-tion of the drug. Furthermore, attention should be paid to the model drug’s low melt-ing point (75–78°C). The application of nanoparticles could be a solution because they are beneficial formulations for Class II drugs of the Biopharmaceutics Classification System (BCS), where the dissolution rate is the rate-limiting step for absorption. The reduction of particle size can increase the dissolution rate as a result of a higher specif-ic surface area. Therefore, the preparation of inhalable IBU nanoparticles with in-creased dissolution may be used as an alternative formulation approach for the tar-geted delivery of IBU to the lungs. [11–13]. There are literature about the micronization and nanonization of IBU, but mostly to enhance its water-solubility and basically for oral application [14–18]. Inhalable formulations for local treatment are published about micro-sized IBU containing particles [19,20] and only one describes nanoparticle ag-glomerates [21].

Our research work aimed to investigate the feasibility of the preparation of an ibuprofen nanosuspension to achieve rapid dissolution. From the nanosuspension we planned to develop inhalable microparticles to achive high lung deposition. Therefore, the combination of various preparation methods was used Additives were applied to investigate the possibility to change the formulations physico-chemical and aerody-namic properties and improve dissolution rate of the drug. Considering the different nanosizing techniques, ultrasound-sonication was chosen. It is a reproducible, cost-effective, and scalable way of preparing nanosuspensions [22–24]. The solidifica-tion of the nanosuspensions combines the advantages of liquid nanosuspensions (i.e. enhanced dissolution and solubility) with the benefits of solid formulations (i.e. stabil-ity, easier applicability) producing nanoparticle agglomerates. Spray-drying is a popu-lar process from an industrial perspective because it is also cost- and time-effective and scalable [25–28]. Preparation of nanosuspensions by sonication, combining nano spray-drying to produce dry powders, is suggested as a formulation approach for “nano-in-micro” particles with enhanced dissolution and aerosolization efficiency [29–33]. The novelty of this work was to formulate suitable nano-embedded-microparticles containing IBU for pulmonary delivery, using modern preparation techniques. Inhala-ble dry powder inhalers containing nanosized IBU could provide a potential local treatment for CF.

  1. Table 1 is vague. Please modify it to clarify the values presented.

Thank you for the comment, the table is clarified.

The final composition of the spray-dried samples and the yield of the nano spray-drying are shown in Table 1.

Table 1. Ratio between the ingredients of the spray-dried samples and physical mixtures.

Sample

IBU (mg)

PVA (mg)

MAN(mg)

LEU (mg)

Yield (%)

IBU_PVA_spd

40

80

-

-

50.00 ± 0.02

IBU_PVA_pm

40

80

-

-

-

IBU_PVA_MAN_LEU_spd

40

80

40

20

46.33 ± 0.08

IBU_PVA_MAN_LEU_pm

40

80

40

20

-

  1. Section 2.2.1. Spray-ultrasound associated solvent diffusion-based nanoprecipitation: was the organic solvent evaporated? Did the authors determine the residue solvent?

Thank you for the comment. The boiling point of the EtOAc is 77 °C, therefore we can assume, that the whole amount of the EtOAc was evaporated during the spray-drying method at 70 °C.  This temperature is close to the boiling point, and the large surface of the small particles provides the surface for evaporation. The residue solvent was not determined yet. According to to ICH Q3C(R6) guideline [1] EtOAC is in Class 3, may be regarded as less toxic and of lower risk to human health. Class 3 includes no solvent known as a human health hazard at levels normally accepted in pharmaceuticals. It is considered that amounts of these residual solvents of 50 mg per day or less (corresponding to 5000 ppm or 0.5%) would be acceptable without justification. Higher amounts may also be acceptable provided they are realistic in relation to manufacturing capability and good manufacturing practice.

  1. Committee for Human Medicinal Products ICH guideline Q3C (R5) on impurities: Guideline for Residual Solvents. Int. Conf. Harmon. Tech. Requir. Regist. Pharm. Hum. Use 2015, 44, 24.
  2. Section 2.2.2. Nano spray-drying: please convert the pump 20% to L/h or mL/min.

Thank you for the comment. Using the device, we can set the pump as a percentage value. The actual flow rate depends on the properties of the fluid and the equipment. The maximum sample throughput according to the manufacturer is 200 mL/h [2]. The feed rate increases with the spray mesh size, the setting of the relative spray rate intensity, the recirculation pump rate, and it depends on the feed formulation (i.e. the substance, the solid concentration, the solvent type, and the addition of surfactant). Generally, a higher solid concentration results in a lower feed rate. The addition of organic solvents and surfactants to the water tends to increase the feed rates, which is due to to the lower surface tensions [3].

  1. BUCHI Pharma & Chemistry BUCHI R & D Solutions.
  2. Arpagaus, C.; Collenberg, A.; Rütti, D.; Assadpour, E.; Jafari, S.M. Nano spray drying for encapsulation of pharmaceuticals. Int. J. Pharm. 2018, 546, 194–214, doi:10.1016/j.ijpharm.2018.05.037.
  3. The authors should include a figure to present the size distribution of the spray-dried powders.

Thank you for the comment. Here you can find the size distribution figures of the spray-dried formulations. Hence the heterodispersity, which is also presented by the Span values, we would not like to insert them in to the main article.

  1. Figure 3: Please explain the peak at 53.86 °C of IBU_PVA_spd.

Thank you for the comment.

After the combined preparation method, the DSC curves showed a lower endothermic peak at 54.86 °C in the case of IBU_PVA_spd. It indicates a decrease in the crystallinity, and the residual crystals melted at a lower temperature because of the particle size reduction of the drug.

  1. The authors should include the yield of the spray-drying process.

The final composition of the spray-dried samples and the yield of the nano spray-drying are shown in Table 1

Table 1. Ratio between the ingredients of the spray-dried samples and physical mixtures.

Sample

IBU (mg)

PVA (mg)

MAN(mg)

LEU (mg)

Yield (%)

IBU_PVA_spd

40

80

-

-

50.00 ± 0.02

IBU_PVA_pm

40

80

-

-

-

IBU_PVA_MAN_LEU_spd

40

80

40

20

46.33 ± 0.08

IBU_PVA_MAN_LEU_pm

40

80

40

20

-

  1. The authors should determine the IBU content in the spray-dried powders and compare them with theoretical values,

Thank you for the comment. We determined the IBU content of the samples, and they were close to the theoretical numbers. Here you can see the results in the following table.

Sample

Theorethical API (%)

Experimental API (%)

IBU_PVA_MAN_LEU_spd

33.33

29.52 ± 0.75

IBU_PVA_MAN_LEU_pm

22.22

19.38 ± 0.62

  1. The authors should clarify the reasons for choosing MAN and LEU as well as their concentrations (1 and 0.5%, respectively) in the spray drying.

Thank you for the comment. We applied mannitol and leucine as additives to improve the aerodynamic properties of the final samples [4],[5]. Mannitol is also helpful in cystic fibrosis, thanks to its osmotic effect, it can dilute the thick mucus [6], [7]. The ratios was based on literature data [8].

  1. Pilcer, G.; Amighi, K. Formulation strategy and use of excipients in pulmonary drug delivery. Int. J. Pharm. 2010, 392, 1–19, doi:10.1016/j.ijpharm.2010.03.017.
  2. Zillen, D.; Beugeling, M.; Hinrichs, W.L.J.; Frijlink, H.W.; Grasmeijer, F. Natural and bioinspired excipients for dry powder inhalation formulations. Curr. Opin. Colloid Interface Sci. 2021, 56, 101497, doi:10.1016/j.cocis.2021.101497.
  3. Jaques, A.; Daviskas, E.; Turton, J.A.; McKay, K.; Cooper, P.; Stirling, R.G.; Robertson, C.F.; Bye, P.T.P.; LeSouëf, P.N.; Shadbolt, B.; et al. Inhaled mannitol improves lung function in cystic fibrosis. Chest 2008, 133, 1388–1396, doi:10.1378/chest.07-2294.
  4. Young, P.M.; Salama, R.O.; Zhu, B.; Phillips, G.; Crapper, J.; Chan, H.K.; Traini, D. Multi-breath dry powder inhaler for delivery of cohesive powders in the treatment of bronchiectasis. Drug Dev. Ind. Pharm. 2015, 41, 859–865, doi:10.3109/03639045.2014.909841.
  5. Malamatari, M.; Somavarapu, S.; Kachrimanis, K.; Buckton, G.; Taylor, K.M.G. Preparation of respirable nanoparticle agglomerates of the low melting and ductile drug ibuprofen: Impact of formulation parameters. Powder Technol. 2017, 308, 123–134, doi:10.1016/j.powtec.2016.12.007.

  1. Please clarify which one is correct, D[0.5] or D[0,5].

Thank you for your comment. The correct form is D[0.5].

Reviewer 2 Report

The article contains interesting material and is devoted to obtaining nanosized particles of the drug ibuprofen for their further use in the form of inhalations for the treatment of pulmonary diseases. The resulting ibuprofen nanoparticles have been characterized in detail by a wide range of physicochemical methods, such as light scattering, laser diffraction, electron microscopy, X-ray powder analysis, UV spectroscopy, and many others. Additionally, using a complex of special methods, the aerodynamic characteristics of the obtained nanoparticles of the drug were studied, and the solubility of this solid form of the drug was evaluated in vitro.

The article is written in a well-read and perceived scientific language, and will be useful for many readers of the journal, both specialists in this field and nonspecialists. I believe that the article can be recommended for publication in the journal after appropriate corrections, which in my opinion would improve the perception of the material obtained by the authors.

The following points should be taken into consideration:

1)     In principle, references to publications in the article cover a wide time range up to modern publications. However, it seemed to me that the Introduction is more devoted to the description of diseases and the mechanism of action of drugs than to the description of the current state of affairs in the development of methods for obtaining nanoparticles of drugs and, in particular, ibuprofen. In this regard, the novelty of the proposed article does not look so obvious. In particular, In the Part 2.2.2. the authors claim that “…The novelty of this work was the combination of various preparation methods, the use of additives to explore the possibility of changing its physico-chemical and aerodynamic properties and improving its rate of dissolution…”. However, many of these issues are considered in sufficient detail in the literature, for example, recent publications (and references in them) [Chishti N, Dehghan MH. Nano-embedded microparticles based dry powder inhaler for lung cancer treatment. J Res Pharm. 2020; 24(3): 425-435, doi:10.35333/jrp.2020.165; Spray Drying for the Preparation of Nanoparticle-Based Drug Formulations as Dry Powders for Inhalation Maria Malamatari, Anastasia Charisi, Stavros Malamataris, Kyriakos Kachrimanis and Ioannis Nikolakakis, Processes 2020, 8, 788; doi:10.3390/pr8070788). Even more significant is the number of publications devoted to obtaining various types of ibuprofen nanoparticles, only the lack of access to databases does not allow me to give at least some of them. I would like to see in the article a more specific definition of the novelty of the proposed approaches in comparison with the known and obtained literature data.

2)     In the Part 2.2.1. of the article the Authors claim that "Preliminary experiments were carried out to select a suitable organic solvent and stabilizer." But further nothing is said about the reasons for choosing EtOAc. And how do the Authors prove that the solvent used does not form complexes with ibuprofen or other components and, in one form or another, does not remain in the final material?

3)     In Figure 2, the scale of the presented diffraction patterns is greatly reduced, which reduces their information content, including for the original PVA sample. The diffraction patterns for IBU_PVA_MAN_LEU_spd and IBU_PVA_spd are not informative and do not allow one to judge the phase state of the samples. It is necessary to bring them on an enlarged scale so that the existing peaks can be observed. Note that the diffraction curve of the IBU_PVA_MAN_LEU_spd sample shows peaks (particularly at about 9.5 degrees 2theta) that cannot be identified as belonging to the original components and may indicate complexation.

4)     According to DSC data (Figure 3), the IBU_PVA_spd sample has crystallinity, while according to X-ray powder data it is completely amorphous. In this case, the appearance of a DSC peak in a region different from the position of the peak for the initial IBU (which is also observed for IBU_PVA_pm) may also indicate the presence of a different crystalline form in the sample. All this requires an explanation.

5)     Are the designations of the samples in Figure 4 mixed up? It seems surprising that the UV spectra for the IBU_PVA and IBU_PVA_MAN_LEU samples practically coincide.

6)     It seems to me that the description of the results of Dynamic light scattering is not the best. It does not become clear from it which version of the samples presented is better or worse, more suitable or not. It's strange - the size less than 200 is good, and the size more than two hundred is even better. So where is the difference and the choice of the best and optimal? All samples have a negative zeta potential, so they are negative for all three, so what?

7)     It follows from the above analysis of the electron microscopy data that the observed particle sizes in the SEM records were 671.41±132.38 nm for IBU_PVA_spd and 561.41±111.62 nm for IBU_PVA_MAN_LEU_spd, while Figure 1 gives the impression that, on the contrary, the particle sizes increase with the addition of components. Might not be the best angle.

8)     There are several grammatical errors and mistakes in the paper, which need to be addressed:

-        on the page 5 (line 196)   - CuKα1 instead of CuKλ1;

-        on the page 5 (line 198) – EVA 28 – possibly wrong program name;

-        on the page 6 (lines 271-272), Table 3 -  Dimentionality of D[0,5] and SSA is missed.

-        Page 11 (lines 351-352), “The total amount of the nano-sized drug released from the spray-dried samples within the first 2min (Figure 6.)” - A sentence without a predicate.

Author Response

The following points should be taken into consideration:

  • In principle, references to publications in the article cover a wide time range up to modern publications. However, it seemed to me that the Introduction is more devoted to the description of diseases and the mechanism of action of drugs than to the description of the current state of affairs in the development of methods for obtaining nanoparticles of drugs and, in particular, ibuprofen. In this regard, the novelty of the proposed article does not look so obvious. In particular, In the Part 2.2.2. the authors claim that “…The novelty of this work was the combination of various preparation methods, the use of additives to explore the possibility of changing its physico-chemical and aerodynamic properties and improving its rate of dissolution…”. However, many of these issues are considered in sufficient detail in the literature, for example, recent publications (and references in them) [Chishti N, Dehghan MH. Nano-embedded microparticles based dry powder inhaler for lung cancer treatment. J Res Pharm. 2020; 24(3): 425-435, doi:10.35333/jrp.2020.165; Spray Drying for the Preparation of Nanoparticle-Based Drug Formulations as Dry Powders for Inhalation Maria Malamatari, Anastasia Charisi, Stavros Malamataris, Kyriakos Kachrimanis and Ioannis Nikolakakis, Processes 2020, 8, 788; doi:10.3390/pr8070788). Even more significant is the number of publications devoted to obtaining various types of ibuprofen nanoparticles, only the lack of access to databases does not allow me to give at least some of them. I would like to see in the article a more specific definition of the novelty of the proposed approaches in comparison with the known and obtained literature data.

Thank you for the comment. The introduction is modified.

The poor-water solubility of ibuprofen causes a certain difficulty in the formula-tion of the drug. Furthermore, attention should be paid to the model drug’s low melt-ing point (75–78°C). The application of nanoparticles could be a solution because they are beneficial formulations for Class II drugs of the Biopharmaceutics Classification System (BCS), where the dissolution rate is the rate-limiting step for absorption. The reduction of particle size can increase the dissolution rate as a result of a higher specif-ic surface area. Therefore, the preparation of inhalable IBU nanoparticles with in-creased dissolution may be used as an alternative formulation approach for the tar-geted delivery of IBU to the lungs. [11–13]. There are literature about the micronization and nanonization of IBU, but mostly to enhance its water-solubility and basically for oral application [14–18]. Inhalable formulations for local treatment are published about micro-sized IBU containing particles [19,20] and only one describes nanoparticle ag-glomerates [21].

Our research work aimed to investigate the feasibility of the preparation of an ibuprofen nanosuspension to achieve rapid dissolution. From the nanosuspension we planned to develop inhalable microparticles to achive high lung deposition. Therefore, the combination of various preparation methods was used Additives were applied to investigate the possibility to change the formulations physico-chemical and aerody-namic properties and improve dissolution rate of the drug. Considering the different nanosizing techniques, ultrasound-sonication was chosen. It is a reproducible, cost-effective, and scalable way of preparing nanosuspensions [22–24]. The solidifica-tion of the nanosuspensions combines the advantages of liquid nanosuspensions (i.e. enhanced dissolution and solubility) with the benefits of solid formulations (i.e. stabil-ity, easier applicability) producing nanoparticle agglomerates. Spray-drying is a popu-lar process from an industrial perspective because it is also cost- and time-effective and scalable [25–28]. Preparation of nanosuspensions by sonication, combining nano spray-drying to produce dry powders, is suggested as a formulation approach for “nano-in-micro” particles with enhanced dissolution and aerosolization efficiency [29–33]. The novelty of this work was to formulate suitable nano-embedded-microparticles containing IBU for pulmonary delivery, using modern preparation techniques. Inhala-ble dry powder inhalers containing nanosized IBU could provide a potential local treatment for CF.

2)   In the Part 2.2.1. of the article the Authors claim that "Preliminary experiments were carried out to select a suitable organic solvent and stabilizer." But further nothing is said about the reasons for choosing EtOAc. And how do the Authors prove that the solvent used does not form complexes with ibuprofen or other components and, in one form or another, does not remain in the final material?

Thank you for the comment. EtOAc was chosen, because it can solve appropriate amount of the IBU, and partially water soluble. Furthermore, based on literature it is applicable for the preparation of nanosuspension using ultrasound associated methods [1], [2]. The boiling point of the EtOAc is 77 °C [3], therefore we can assume, that the whole amount of EtOAc was evaporated during the spray-drying method at 70 °C.  This temperature is close to the boiling point, and the large surface of the small particles provides the surface for evaporation. Probably, during this short period of efficient drying, formation of complexes (inclusion complexes) does not occur. The residue solvent was not determined yet. According to to ICH Q3C(R6) guideline EtOAC is in Class 3, may be regarded as less toxic and of lower risk to human health. Class 3 includes no solvent known as a human health hazard at levels normally accepted in pharmaceuticals. It is considered that amounts of these residual solvents of 50 mg per day or less (corresponding to 5000 ppm or 0.5%) would be acceptable without justification. Higher amounts may also be acceptable provided they are realistic in relation to manufacturing capability and good manufacturing practice [4].

  1. Kocbek, P.; Baumgartner, S.; Kristl, J. Preparation and evaluation of nanosuspensions for enhancing the dissolution of poorly soluble drugs. Int. J. Pharm. 2006, 312, 179–186, doi:10.1016/j.ijpharm.2006.01.008.
  2. Ambrus, R.; Kocbek, P.; Kristl, J.; Šibanc, R.; Rajkó, R.; Szabó-Révész, P. Investigation of preparation parameters to improve the dissolution of poorly water-soluble meloxicam. Int. J. Pharm. 2009, 381, 153–159, doi:10.1016/j.ijpharm.2009.07.009.
  3. Kenig, E.Y.; Bäder, H.; Górak, A.; Beßling, B.; Adrian, T.; Schoenmakers, H. Investigation of ethyl acetate reactive distillation process. Chem. Eng. Sci. 2001, 56, 6185–6193, doi:10.1016/S0009-2509(01)00206-8.
  4. Committee for Human Medicinal Products ICH guideline Q3C (R5) on impurities: Guideline for Residual Solvents. Int. Conf. Harmon. Tech. Requir. Regist. Pharm. Hum. Use 2015, 44, 24.

3)   In Figure 2, the scale of the presented diffraction patterns is greatly reduced, which reduces their information content, including for the original PVA sample. The diffraction patterns for IBU_PVA_MAN_LEU_spd and IBU_PVA_spd are not informative and do not allow one to judge the phase state of the samples. It is necessary to bring them on an enlarged scale so that the existing peaks can be observed. Note that the diffraction curve of the IBU_PVA_MAN_LEU_spd sample shows peaks (particularly at about 9.5 degrees 2theta) that cannot be identified as belonging to the original components and may indicate complexation.

Thank you for the comment. On the following magnified XRPD spectra, we can observe the polymorph transformation of MAN. It explains the peak at about 9.5 2theta degrees [5][6][7].

  1. Cornel, J.; Kidambi, P.; Mazzotti, M. Precipitation and transformation of the three polymorphs of d-mannitol. Ind. Eng. Chem. Res. 2010, 49, 5854–5862, doi:10.1021/ie9019616.
  2. Yoshinari, T.; Forbes, R.T.; York, P.; Kawashima, Y. Moisture induced polymorphic transition of mannitol and its morphological transformation. Int. J. Pharm. 2002, 247, 69–77, doi:10.1016/S0378-5173(02)00380-0.
  3. Smith, R.R.; Shah, U. V.; Parambil, J. V.; Burnett, D.J.; Thielmann, F.; Heng, J.Y.Y. The Effect of Polymorphism on Surface Energetics of D-Mannitol Polymorphs. AAPS J. 2017, 19, 103–109, doi:10.1208/s12248-016-9978-y.

4)    According to DSC data (Figure 3), the IBU_PVA_spd sample has crystallinity, while according to X-ray powder data it is completely amorphous. In this case, the appearance of a DSC peak in a region different from the position of the peak for the initial IBU (which is also observed for IBU_PVA_pm) may also indicate the presence of a different crystalline form in the sample. All this requires an explanation.

Thank you for your comment. After the combined preparation method, the DSC curves showed a lower endothermic peak at 54.86 °C in the case of IBU_PVA_spd. It indicates a decrease in the crystallinity, and the residual crystals melted at a lower temperature because of the particle size reduction of the drug. On the XRPD spectra we can observe a small characteristic peak of IBU, which shows the residual crystal form of IBU. You can see it on the following magnified figure.

5)   Are the designations of the samples in Figure 4 mixed up? It seems surprising that the UV spectra for the IBU_PVA and IBU_PVA_MAN_LEU samples practically coincide.

Thank you for the comment. The signs are correct.

6)   It seems to me that the description of the results of Dynamic light scattering is not the best. It does not become clear from it which version of the samples presented is better or worse, more suitable or not. It's strange - the size less than 200 is good, and the size more than two hundred is even better. So where is the difference and the choice of the best and optimal? All samples have a negative zeta potential, so they are negative for all three, so what?

Thank you for the comment.

The results of the DLS investigations are presented in Table 2. In case of the IBU suspension, the particle size of the drug decreased into the nano size range (178.72±7.73 nm) after the ultrasound-associated precipitation. The average mesh space range of CF sputum is about 60− 300 nm, therefore the prepared IBU nanoparticles may go through the dense fiber mesh of CF mucus. The particle size lower than 200 nm and the negative ζ-potential will probably lead to high biocompatibility in human bronchial epithelium cells. [32]. According to the ζ-potential results (-20.91±2.10 mV), the nanosuspension was described as a stable suspension system. The stability results supported that, thus the particle size and zeta potential remained persistent 7 days long. [30], [31].

The diameters of the dispersed nano spray-dried particles, which were produced from the IBU nanosuspension, were between 300-400 nm. It predicts, that the dry particles will disintegrate into nanoparticles, when they deposit the lung fluid. Particles with a mean particle diameter between 200 and 500 nm are beneficial to reaching increased dissolution and cellular uptake [33]. The negative ζ-potential values of the spray-dried formulations mean that the systems are more degradable and therefore less retentive in the airways. It is advantageous in CF because the particles will not induce further inflammation, infection, and fibrosis [34].

7)   It follows from the above analysis of the electron microscopy data that the observed particle sizes in the SEM records were 671.41±132.38 nm for IBU_PVA_spd and 561.41±111.62 nm for IBU_PVA_MAN_LEU_spd, while Figure 1 gives the impression that, on the contrary, the particle sizes increase with the addition of components. Might not be the best angle.

Thank you for the comment. In the case of IBU_PVA_spd, a broader particle size distribution was observed, therefore on Figure 1. we can see also smaller and larger particles. The average particle size is larger, than in case of IBU_PVA_MAN_LEU_spd according to the ImageJ analysis, which gives accurate results.

8)   There are several grammatical errors and mistakes in the paper, which need to be addressed:

Thank you for the comment. The text is corrected.

-on the page 5 (line 196) - CuKα1 instead of CuKλ1;

The radiation source was Cu Kα1 radiation (α=1.5406 Å).

-on the page 5 (line 198) – EVA 28 – possibly wrong program name;

The data were evaluated with DIFFRACTplus EVA software (Bruker AXS GmbH, Karlsruhe, Germany).

-on the page 6 (lines 271-272), Table 3 -Dimentionality of D[0,5] and SSA is missed.

Table 3. Laser diffraction results of the initial API and the spray-dried formulations.

Sample

D[0.5] (μm)

Span

SSA (m2/g)

IBU_raw

29.33±0.94

0.74±0.08

0.25±0.04

IBU_PVA_spd

2.13±0.10

11.89±2.67

4.93±0.11

IBU_PVA_MAN_LEU_spd

1.57±0.04

5.55±0.64

5.36±0.20

-Page 11 (lines 351-352), “The total amount of the nano-sized drug released from the spray-dried samples within the first 2min (Figure 6.)” - A sentence without a predicate.

The total amount of the nano-sized drug was released from the spray-dried samples within the first 2 min (Figure 6.).

Round 2

Reviewer 1 Report

The manuscript was appropriately revised. However, there are still several issues to consider, as follows.

1. Introduction: The authors only placed citations at the end of paragraphs #1-5. This is inappropriate. Please modify them.

2. It is OK not to include the figures presenting the size distribution of the spray-dried powders in the main text. However, the authors should include them in Supplementary materials.

3. Determination of the IBU content in the spray-dried powders and comparison with theoretical values: the authors responded to the reviewer's comment. However, the authors should include the results and discussion in the main text to clarify to readers.

4. The reasons for choosing MAN and LEU as well as their concentrations (1 and 0.5%, respectively) in the spray drying:  the authors responded to the reviewer's comment. However, the authors should include the answer in the main text to clarify to readers.

5. The authors still used D[0,5] (in Table 3), although they mentioned that D[0.5] must be used.

Author Response

Respone to Reviewer1

The manuscript was appropriately revised. However, there are still several issues to consider, as follows.

  1. Introduction: The authors only placed citations at the end of paragraphs #1-5. This is inappropriate. Please modify them.

Thank you for your comment, the citations were modified.

  1. It is OK not to include the figures presenting the size distribution of the spray-dried powders in the main text. However, the authors should include them in Supplementary materials.

Thank you for your comment. The figures were added to the supplementary materials.

Supplementary Materials: The following supporting information can be downloaded at: www.mdpi.com/xxx/s1, Figure S1: particle size distribution of IBU_PVA_spd, Figure S2: particle size distribution of IBU_PVA_MAN_LEU_spd.

  1. Determination of the IBU content in the spray-dried powders and comparison with theoretical values: the authors responded to the reviewer's comment. However, the authors should include the results and discussion in the main text to clarify to readers.

Thank you for your comment. The text was modified.

The IBU content of the spray dried samples was determined. The results were close to the theoretical values.

  1. The reasons for choosing MAN and LEU as well as their concentrations (1 and 0.5%, respectively) in the spray drying:  the authors responded to the reviewer's comment. However, the authors should include the answer in the main text to clarify to readers.

Thank you for your comment. The text was modified.

The ratios was based on literature data [1].

  1. Malamatari, M.; Somavarapu, S.; Kachrimanis, K.; Buckton, G.; Taylor, K.M.G. Preparation of respirable nanoparticle agglomerates of the low melting and ductile drug ibuprofen: Impact of formulation parameters. Powder Technol. 2017, 308, 123–134, doi:10.1016/j.powtec.2016.12.007.
  2. The authors still used D[0,5] (in Table 3), although they mentioned that D[0.5] must be used.

Thank you for your comment. The table was modified.

Table 3. Laser diffraction results of the initial API and the spray-dried formulations.

Sample

D[0.5] (μm)

Span

SSA (m2/g)

IBU_raw

29.33±0.94

0.74±0.08

0.25±0.04

IBU_PVA_spd

2.13±0.10

11.89±2.67

4.93±0.11

IBU_PVA_MAN_LEU_spd

1.57±0.04

5.55±0.64

5.36±0.20

*Data are means ± SD (n = 3 independent measurements).

Round 3

Reviewer 1 Report

The manuscript was appropriately revised and can be accepted as is.